# Entity-Controlled Synthetic Text Generation using Contextual Question and Answering with Pre-trained Language Models

**Karan Aggarwal**
Amazon
Seattle, WA
kagg@amazon.com

**Henry Jin**[*]
Harvard University
Cambridge, MA
helinjin@g.harvard.edu

**Aitzaz Ahmad**
Amazon
Seattle, WA
aitzaza@amazon.com

## Abstract

Recent advancements in Natural Language Processing (NLP) algorithms have resulted in state-of-the-art performance on Named Entity Recognition (NER) tasks. These algorithms typically require high-quality labeled datasets for training models. However, training NLP models effectively can suffer from issues such as scarcity of labeled data, data bias and under-representation, and privacy concerns with using sensitive data for training. Generating synthetic data to train models is a promising solution to mitigate these problems. We propose a contextual question and answering approach using pre-trained language models to synthetically generate entity-controlled text. Entity-controlled text generation is then used to augment small labeled datasets for downstream NER tasks. We evaluate this proposed method on two publicly available datasets, and measure the quality of generated texts quantitatively. We find that the model is capable of producing full text samples with the desired entities appearing in a stochastically controllable way, while retaining sentence coherence closest to the real world data. Evaluations on downstream NER tasks show significant improvements in low-labeled data regime, and in using purely synthetic data for NER to alleviate privacy concerns.

## 1 Introduction

Many tasks in NLP require large amounts of high-quality labeled data to train sufficiently accurate and useful models. However, in many domains, such as finance and healthcare, access to labeled data is often limited. In these domains, annotating data often requires strong domain expertise and therefore, crowdsourcing of labeled data is infeasible. The cost of annotating data by training an expert workforce is often too high for feasibility. Even if it were financially feasible to annotate data, there are concerns with using customer data for training large language models, and potentially endangering customer privacy.

Recent studies have raised concerns about leakage of training data (potentially sensitive information) from the trained language models [2, 9, 11]. A small collection of labeled data also runs the risk of bias creeping in the data and may result in algorithms and models that reflect or even exploit this

---

[*]This work was done during Henry's intership at Amazon

NeurIPS 2022 Workshop on Synthetic Data for Empowering ML Research, New Orleans, LA.

inherent bias. It also degrades the capability of models to generalize since they may have been trained on a small dataset where certain population groups or patterns were under-represented [19, 28, 27, 7]. These issues demand solutions that can perform well in low-labeled data regimes and can combat privacy concerns and data bias.

Synthetic data generation presents a promising solution to address the issues outlined above [1, 5]. By generating data synthetically, we can augment small labeled datasets to build a training set large enough for efficient learning of large models. Synthetic data generation also promotes privacy by hiding sensitive customer data. Synthetic data generation can also reduce bias in the data by conditionally generating it in a way that all population groups are sufficiently represented. In particular, the field of conditional or controlled synthetic text generation has received increased attention in recent years. Controlled text generation provides the ability to control for traits such as tone, formality, sentiment, and topic in the generation of a language model [23, 26]. This lends controlled synthetic text generation as a useful technique for augmenting small or privacy-sensitive datasets. However, there has been limited work on the topic of entity-controlled synthetic text generation, i.e., the task of generating coherent text while controlling for the named entities that appear in the generation [8].

In this paper, we study the problem of entity-controlled synthetic text generation. We propose a Contextual Question Answering based pre-trained language model that can produce coherent text which contains specific entity tokens, generated in an order provided by the user. We are motivated by the need to synthetically augment datasets to improve performance on downstream NER tasks. Our contributions in this work are as follows. 1) We propose a contextual question and answering approach using pre-trained language models to generate entity-controllable blocks of text, which can be chained to produce full training text samples, 2) Our method is capable of generating coherent texts that beat the baseline methods in grammaticality and distinctness metrics, and 3) Evaluations on publicly available NER datasets show a significant improvement in performance in low-labeled data regimes, and for scenarios consisting of purely synthetic training data to promote privacy.

## 2   Related Work

There has been limited work in the area of entity-controlled text generation. We can group prior works into: Controlled text generation, data-to-text generation, and entity-controlled text generation.

**Controlled text generation**   These methods have been designed to control certain aspects of generated text [25, 3, 16] like sentiment [23] or concepts [26]. Furthermore, they have been used in summarization [20] tasks as well. These methods however, mostly focus on changing one aspect of the generated text like a topic or sentiment. Our goal is to generate a coherent text based on the set of entities we want to appear in the text.

**Data-to-text generation**   This is a well studied class of problems where the idea is to convert a given set of words or structured data from tables into a piece of text. Most popular problem is table summary generation, also called table-to-text [14, 15, 4] or keyword to text methods [16, 21]. While similar to our problem, the key difference is that they have a pre-defined set of words that just need to appear in every generated text while we can have a variable number of entities that appear in each generated text. For example, table-to-text methods would be given a set of four words per row belonging to one entity type, while we need to generate any number of entities (even repeated) as they occur in most real world corpus.

**Entity-controlled generation**   To the best of our knowledge, only Dong et al. [8] have worked on this problem. They use a two pronged approach to generate text with given entity types and their mentions. The entity predictor, is used for generating an entity tag, indicating that an entity must be injected at a particular location in the text. When these tags are generated, the desired entity type is injected and the second stage using the mention predictor generates the specific entity mention (word instantiation) that belongs to the entity tag. They use a RNN based sequence-to-sequence architecture to achieve this. We make a comparison with their method and found that their method generated repetitive text, and does not generate as realistic text as ours. Additionally, they do not make a comparison on a downstream task, while we present an analysis on downstream NER task.

# 3 Methodology

Our methodology uses a pre-trained language model and a question and answering training approach to generate blocks of text with desired entity tokens. This approach is able to reliably generate augmented text samples while retaining sentence coherence. Our method builds off the work of Dong et al. by training on blocks of text and chaining such blocks to generate text samples. In addition, we opt to use a pretrained transformer-based language model in place of a recurrent network to take advantage of the benefits of using finetuned and pre-trained language models. We expect that using the pre-trained model helps in diversity of the generated text.

## 3.1 Training

To use the approach of question and answering, we first preprocess our real world training text samples into blocks, whereby each block is composed of non-entity tags and ends with an entity token. Every text sample is then decomposed into these blocks of text. In addition to the entity tokens in the training corpus, an end of text token is also added to the end of every text sample. Therefore, a full text sample generation consists of chaining generated blocks until a block with an <ENDTEXT> token appears.

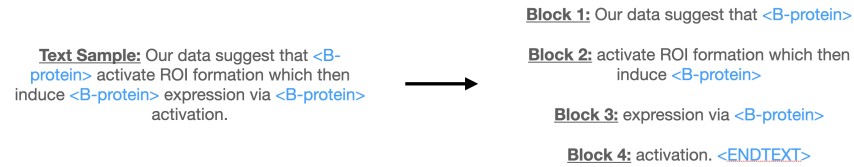

Figure 1: This figure depicts how a text sample with three <B-protein> entities is processed into blocks that conclude with an entity for training with our Question and Answering approach. An <ENDTEXT> token always defines the final block of a decomposed text sample.

After decomposing text samples into such blocks, we arrange blocks into the question and answering format, which consists of three segments: context, question and answer. The context segment provides preceding text blocks, the question segment prompts the model for the desired token, and the answer block is the desired generation.

The context section consists of all blocks preceding a particular block belonging to the same text sample. This was motivated by the need for the model to be aware of the context for a particular generation. The generation of each block must be a continuation of preceding blocks in order for sentence level coherence to be maintained. Consequently, training the model on preceding blocks, provides a signal for the text's context in order for the generation in the answer segment to be a seamless continuation of the preceding blocks.

The question segment prompts the model for the desired entity to appear in the next block, and is therefore the mechanism by which we control for the desired entity token to be generated. Following the "Question: " tag is a single token representing the desired entity.

The answer segment contains the desired text block to be generated. The final token in this block will therefore be the same token as in the question segment. With this three segment format, every block from the corpus represents a training sample for the language model. Figure 2 illustrates this three segment structure of each training sample.

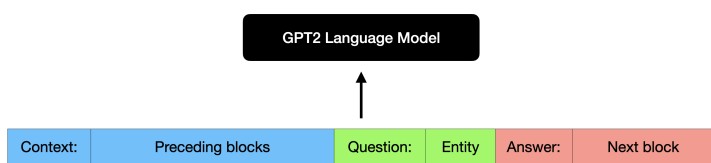

Figure 2: This figure illustrates the segmentation format of each training sample into the language model.

**Context:** Our data suggest that <B-protein> activate ROI formation which then induce <B-protein>

**Question:** <B-protein>

**Answer:** expression via <B-protein>

Figure 3: An example for how a training sample is generated for Block 3 of the example text shown in Figure 1.

## 3.2 Generation during Inference

At inference time, the language model generates text conditioned on the first two segments of context and question. To generate the first block of a text, the context segment is blank, while the question segment contains the desired token to be generated in the first block. The model then completes the answer segment with a generated block, which is inserted into the context segment for the next block generation. A full text sample then is produced by concatenating blocks until an <ENDTEXT> token.

## 3.3 Metrics

To evaluate the model, we quantitatively measure the quality of generation and then measure the performance on NER task using precision, recall, and $F_1$ scores. We use three generation quality metrics that have been used in the prior works [8]. **Grammaticality** [24] measures the grammaticality probability evaluated with a Roberta CoLa grammaticality model. **Perplexity** measures the 'surprisingness' of the generated text evaluated on a GPT model [17]. **Distinctness** [12] measures the uniqueness of trigrams in the corpus. **Rouge-L** [13]: One trivial sanity check is regurgitation., *i.e.,* if the generation model is simply memorizing the training set. Rouge-L score measures the similarity of the generated text with the training data by calculating the longest common substrings. Ideally, if the model is not just spitting out the training examples, Rouge-L score should be low.

Table 1: Dataset Statistics: Size of train, validation, test sets and number of entities.

| Dataset | #Training | #Val | #Test | Avg. #Words | #Entities | Description |
|---|---|---|---|---|---|---|
| **JNLPBA** | 18606 | 1938 | 4259 | 22.97 | 5 | Biomedical dataset from GENIA corpus [10] |
| **BC5CDR** | 4561 | 4582 | 4798 | 24.93 | 2 | Biochemical dataset of 1500 PubMed articles containing chemical-disease interactions |

# 4 Experiments

We evaluate our model on two public datasets described in Table 1. We use the following baselines to compare our method with the following three baselines:

**Original Data:** Refers to BERT model trained using the real world training data.

**Random Swap [22]:** We randomly swap entity mentions across the corpus following Vakili et al. [22], e.g., for the entity 'B-DNA', we replace with another randomly chosen mention of 'B-DNA' in our training data. Since, we are following a similar procedure to instantiate a generated entity post generation in our model, we use this as a simple baseline. Additionally, it acts as a sanity check during augmentation, as the training corpus has not changed much except reshuffling of entity mentions.

**EntInjection [8]:** We use the Entity Injection work by Dong et al [8] as that is the only relevant work closest to our work.

Please note, while we did not make a comparison with other Seq2Seq models or just a pure GPT-2 based model without our Q&A framework, EntInjection [8] makes thorough comparisons with such models and showed the generation quality was much worse for those baselines. Using a Q&A framework as we do or the block-by-block generation approach used by EntInjection, allows for a longer text generation, unlike pure Seq2Seq models that are unable to do so.

Table 2: Generation Quality Metrics for the two datasets: Grammaticality, Perplexity, and Distinctness-3 (tri-gram). Cells with best score are highlighted in blue among the three data generation methods. (↑): higher the better; (↓): lower the better.

| | JNPBLA (10%) | | | | BC5CDR (10%) | | | |
| Metric | Original Data (Oracle) | Random Swap [22] | EntInjection [8] | Ours | Original Data (Oracle) | Random Swap [22] | EntInjection [8] | Ours |
|---|---|---|---|---|---|---|---|---|
| Grammaticality(↑) | 0.85 | 0.63 | 0.32 | 0.57 | 0.82 | 0.15 | 0.29 | 0.51 |
| Perplexity(↓) | 400.36 | 605.75 | 796.5 | 488.56 | 388.42 | 5856.61 | 1521.3 | 477.66 |
| Distinctness-3(↑) | 0.74 | 0.82 | 0.2 | 0.58 | 0.72 | 0.92 | 0.06 | 0.59 |
| Rouge-L(↓) | 1.0 | 0.72 | 0.30 | 0.20 | 1.0 | 0.83 | 0.26 | 0.21 |

## 4.1 Experimental Settings

We use the training, validation, and testing data splits provided publicly in the datasets on Huggingface[2]. We use the training dataset (and its mentioned subsets) for training both the text generation models as well as training the downstream NER model. We use BERT [6] for downstream NER task. We measure the generation quality metrics on the generated text from the training dataset. NER results are reported on the complete test set for both the datasets.

For our pre-trained language model, we use an instance of OpenAI's GPT-2 [18]. The model is trained with the Adam optimizer on a learning rate of 1e-3, one hundred warmup steps, and an epsilon of 1e-8. The default CrossEntropy loss function is used, and the model is trained for up to 100 epochs. For the NER task, we train the BERT model for upto 10 epochs with a learning rate of 2e-3. These parameters were set based on hyper-parameter tuning on the validation set. *During generation, we exactly mimic the entity distribution of training data samples.*

# 5 Results and Discussion

## 5.1 Generation Quality

Generation quality results are shown in Table 2 measured on Grammaticality, Perplexity, Distinctness-trigram. We clearly observe that our method is lower on all three metrics against the original dataset, which is expected as ours is synthetically generated data. However, our method works better than the baseline EntInjection [8] on all three metrics across both the datasets. Particularly, for the BC5CDR dataset, we observed that EntInjection tends to generate repetitive text. *The correct benchmark is the random swap as our method inserts the entitities in the same fashion.* We observe for the random swap baseline, distinctness is highest, as expected as we have swapped commonly occurring trigram entities, while the perplexity and grammaticality are worse than all the methods. This shows that random swapping affects the lexical meaning of the text. While we also insert randomly chosen entities in our generated text, these results indicate that our method generates coherent generic text where semantic meaning of the type of the entity is preserved, unlike other baselines.

Our generated data has one of the lowest Rouge-L scores across the two datasets. Hence, our generated data is not simply memorizing the training data and is quite different than the original training data. We can see the huge gap between the generated data through random swapped entities and our generated data; while the former is practically same as the training data, ours is distinct. This is quite important for privacy, as this ensures that an adversary would not be able to extract the original training data trivially from the generated data. Based on these metrics, *we can claim that generated text is semantically closest to the original corpus for all the datasets, while being distinct.*

## 5.2 Named Entity Recognition Task

We took two subsets of the JNLPBA and BC5CDR datasets: 1% and 10% as we found that the performance on datasets was already saturated at their full sizes as number of samples was enough. We didn't find any difference in performance between BERT model trained on the original training dataset, synthetically generated dataset, and augmented dataset with our generated examples. Hence, we present the results on first 1% and 10% examples of training datasets to show the comparisons. We present two settings: (a) w/o augmentation with original training data; and (b) augmentation with original training data. Idea for (a) is to test for privacy-preserving training, and (b) tests effectiveness of the generated data for data augmentation purposes. Generated text is same size as the training set.

---

[2]https://huggingface.co/

Table 3: Precision (P), Recall (R), and $F_1$ scores on NER. Dataset is highlighted in gray, cells with highest and second highest $F_1$ scores for an entity are highlighted in blue and underlined respectively. $\Delta$ is absolute difference in $F_1$ scores of original data and Ours (w/ augmentation).

| | W/o Augmentation | | | | | | | | | | | | W/ Augmentation | | | | | | | | | $\Delta$ |
| Training Data $(\rightarrow)$ | Original Data | | | Random Swap [22] | | | EntInjection [8] | | | Ours | | | Random Swap [22] | | | EntInjection [8] | | | Ours | | | |
| Entity $(\downarrow)$ | P | R | F1 | P | R | F1 | P | R | F1 | P | R | F1 | P | R | F1 | P | R | F1 | P | R | F1 | F1 |
|---|---|---|---|---|---|---|---|---|---|---|---|---|---|---|---|---|---|---|---|---|---|---|
| **JNLPBA (1%)** [Training Samples = 186] | | | | | | | | | | | | | | | | | | | | | | |
| B-DNA | 0.50 | 0.00 | 0.00 | 0.00 | 0.00 | 0.00 | 0.00 | 0.00 | 0.00 | 0.00 | 0.00 | 0.00 | 0.24 | 0.05 | 0.08 | 0.11 | 0.00 | 0.01 | 0.55 | 0.46 | 0.51 | 0.51 |
| B-RNA | 0.00 | 0.00 | 0.00 | 0.00 | 0.00 | 0.00 | 0.00 | 0.00 | 0.00 | 0.00 | 0.00 | 0.00 | 0.10 | 0.03 | 0.05 | 0.00 | 0.00 | 0.00 | 0.51 | 0.25 | 0.33 | 0.33 |
| B-cell-line | 0.00 | 0.00 | 0.00 | 0.00 | 0.00 | 0.00 | 0.00 | 0.00 | 0.00 | 0.00 | 0.00 | 0.00 | 0.33 | 0.01 | 0.01 | 0.00 | 0.00 | 0.00 | 0.39 | 0.05 | 0.09 | 0.09 |
| B-cell-type | 0.52 | 0.39 | 0.45 | 0.00 | 0.00 | 0.00 | 0.00 | 0.00 | 0.00 | 0.79 | 0.20 | 0.32 | 0.45 | 0.39 | 0.42 | 0.39 | 0.50 | 0.44 | 0.62 | 0.63 | 0.63 | 0.18 |
| B-protein | 0.43 | 0.78 | 0.56 | 0.35 | 0.75 | 0.48 | 0.00 | 0.00 | 0.00 | 0.46 | 0.66 | 0.54 | 0.48 | 0.73 | 0.58 | 0.41 | 0.82 | 0.54 | 0.62 | 0.70 | 0.66 | 0.10 |
| I-DNA | 0.87 | 0.13 | 0.23 | 0.62 | 0.01 | 0.02 | 0.00 | 0.00 | 0.00 | 0.50 | 0.34 | 0.40 | 0.44 | 0.56 | 0.49 | 0.49 | 0.39 | 0.43 | 0.62 | 0.67 | 0.65 | 0.42 |
| I-RNA | 0.00 | 0.00 | 0.00 | 0.00 | 0.00 | 0.00 | 0.00 | 0.00 | 0.00 | 0.82 | 0.17 | 0.28 | 0.64 | 0.40 | 0.49 | 0.00 | 0.00 | 0.00 | 0.62 | 0.55 | 0.58 | 0.58 |
| I-cell-line | 0.51 | 0.18 | 0.27 | 0.00 | 0.00 | 0.00 | 0.00 | 0.00 | 0.00 | 0.00 | 0.00 | 0.00 | 0.34 | 0.13 | 0.19 | 0.16 | 0.02 | 0.04 | 0.42 | 0.08 | 0.13 | -0. 14 |
| I-cell-type | 0.69 | 0.40 | 0.51 | 0.59 | 0.09 | 0.15 | 0.00 | 0.00 | 0.00 | 0.53 | 0.74 | 0.62 | 0.60 | 0.52 | 0.56 | 0.63 | 0.41 | 0.49 | 0.62 | 0.73 | 0.67 | 0.14 |
| I-protein | 0.44 | 0.59 | 0.51 | 0.34 | 0.32 | 0.33 | 0.00 | 0.00 | 0.00 | 0.50 | 0.74 | 0.59 | 0.62 | 0.47 | 0.53 | 0.46 | 0.64 | 0.53 | 0.67 | 0.70 | 0.68 | 0.16 |
| Macro Avg. | 0.45 | 0.31 | 0.31 | 0.26 | 0.19 | 0.17 | 0.07 | 0.09 | 0.08 | 0.41 | 0.34 | 0.34 | 0.47 | 0.38 | 0.39 | 0.33 | 0.34 | 0.31 | 0.60 | 0.53 | 0.54 | 0.23 |
| **JNLPBA (10%)** [Training Samples = 1860] | | | | | | | | | | | | | | | | | | | | | | |
| B-DNA | 0.66 | 0.75 | 0.70 | 0.59 | 0.27 | 0.37 | 0.15 | 0.04 | 0.06 | 0.70 | 0.60 | 0.65 | 0.65 | 0.76 | 0.70 | 0.48 | 0.46 | 0.47 | 0.68 | 0.75 | 0.71 | 0.01 |
| B-RNA | 0.60 | 0.81 | 0.69 | 0.21 | 0.03 | 0.06 | 0.00 | 0.00 | 0.00 | 0.67 | 0.61 | 0.64 | 0.65 | 0.75 | 0.70 | 0.50 | 0.01 | 0.02 | 0.63 | 0.78 | 0.70 | 0.01 |
| B-cell-line | 0.41 | 0.70 | 0.51 | 0.30 | 0.26 | 0.28 | 0.38 | 0.03 | 0.06 | 0.42 | 0.51 | 0.46 | 0.39 | 0.70 | 0.50 | 0.31 | 0.08 | 0.13 | 0.41 | 0.67 | 0.51 | 0.00 |
| B-cell-type | 0.79 | 0.61 | 0.69 | 0.57 | 0.29 | 0.39 | 0.28 | 0.09 | 0.14 | 0.79 | 0.47 | 0.59 | 0.76 | 0.60 | 0.67 | 0.50 | 0.58 | 0.54 | 0.79 | 0.63 | 0.70 | 0.01 |
| B-protein | 0.70 | 0.83 | 0.76 | 0.58 | 0.64 | 0.61 | 0.51 | 0.40 | 0.45 | 0.69 | 0.59 | 0.64 | 0.66 | 0.85 | 0.74 | 0.57 | 0.72 | 0.64 | 0.69 | 0.82 | 0.75 | -0.01 |
| I-DNA | 0.73 | 0.84 | 0.78 | 0.65 | 0.15 | 0.25 | 0.17 | 0.09 | 0.12 | 0.66 | 0.70 | 0.68 | 0.73 | 0.82 | 0.77 | 0.54 | 0.63 | 0.58 | 0.72 | 0.86 | 0.79 | 0.01 |
| I-RNA | 0.77 | 0.89 | 0.82 | 0.87 | 0.18 | 0.29 | 0.00 | 0.00 | 0.00 | 0.68 | 0.72 | 0.70 | 0.77 | 0.87 | 0.81 | 0.82 | 0.35 | 0.49 | 0.78 | 0.86 | 0.81 | -0.01 |
| I-cell-line | 0.43 | 0.77 | 0.55 | 0.44 | 0.16 | 0.23 | 0.17 | 0.03 | 0.05 | 0.37 | 0.61 | 0.46 | 0.40 | 0.79 | 0.53 | 0.35 | 0.20 | 0.25 | 0.41 | 0.76 | 0.53 | -0.02 |
| I-cell-type | 0.80 | 0.60 | 0.69 | 0.68 | 0.21 | 0.32 | 0.36 | 0.21 | 0.27 | 0.79 | 0.52 | 0.63 | 0.83 | 0.54 | 0.66 | 0.54 | 0.65 | 0.59 | 0.85 | 0.61 | 0.71 | 0.02 |
| I-protein | 0.79 | 0.76 | 0.77 | 0.66 | 0.28 | 0.39 | 0.35 | 0.28 | 0.31 | 0.68 | 0.69 | 0.69 | 0.76 | 0.75 | 0.76 | 0.66 | 0.59 | 0.62 | 0.76 | 0.78 | 0.77 | 0.00 |
| Macro Avg. | 0.70 | 0.77 | 0.72 | 0.59 | 0.31 | 0.37 | 0.29 | 0.19 | 0.21 | 0.67 | 0.63 | 0.64 | 0.69 | 0.76 | 0.71 | 0.57 | 0.47 | 0.48 | 0.70 | 0.77 | 0.72 | 0.00 |
| **BC5CDR (1%)** [Training Samples = 45] | | | | | | | | | | | | | | | | | | | | | | |
| Disease-B | 0.53 | 0.01 | 0.01 | 0.12 | 0.02 | 0.03 | 0.00 | 0.00 | 0.00 | 0.00 | 0.00 | 0.00 | 0.30 | 0.21 | 0.25 | 0.32 | 0.09 | 0.15 | 0.33 | 0.34 | 0.33 | 0.32 |
| Disease-I | 0.00 | 0.00 | 0.00 | 0.00 | 0.00 | 0.00 | 0.00 | 0.00 | 0.00 | 0.18 | 0.00 | 0.00 | 0.33 | 0.01 | 0.01 | 0.24 | 0.03 | 0.05 | 0.26 | 0.04 | 0.07 | 0.07 |
| Chemical-B | 0.25 | 0.00 | 0.00 | 0.48 | 0.03 | 0.05 | 0.00 | 0.00 | 0.00 | 0.59 | 0.00 | 0.01 | 0.51 | 0.39 | 0.44 | 0.36 | 0.09 | 0.15 | 0.61 | 0.55 | 0.58 | 0.58 |
| Chemical-I | 0.00 | 0.00 | 0.00 | 0.00 | 0.00 | 0.00 | 0.00 | 0.00 | 0.00 | 0.81 | 0.29 | 0.43 | 0.00 | 0.00 | 0.00 | 0.00 | 0.00 | 0.00 | 0.79 | 0.22 | 0.35 | 0.35 |
| Macro Avg. | 0.33 | 0.20 | 0.19 | 0.30 | 0.21 | 0.20 | 0.18 | 0.20 | 0.19 | 0.49 | 0.26 | 0.28 | 0.41 | 0.32 | 0.33 | 0.36 | 0.24 | 0.26 | 0.58 | 0.42 | 0.46 | 0.27 |
| **BC5CDR (10%)** [Training Samples = 456] | | | | | | | | | | | | | | | | | | | | | | |
| Disease-B | 0.71 | 0.69 | 0.70 | 0.48 | 0.50 | 0.49 | 0.31 | 0.59 | 0.41 | 0.70 | 0.62 | 0.66 | 0.56 | 0.72 | 0.63 | 0.62 | 0.75 | 0.68 | 0.71 | 0.68 | 0.70 | 0.00 |
| Disease-I | 0.66 | 0.61 | 0.63 | 0.52 | 0.22 | 0.30 | 0.14 | 0.04 | 0.06 | 0.64 | 0.60 | 0.62 | 0.59 | 0.60 | 0.60 | 0.61 | 0.67 | 0.64 | 0.69 | 0.59 | 0.64 | 0.01 |
| Chemical-B | 0.79 | 0.79 | 0.79 | 0.59 | 0.78 | 0.67 | 0.63 | 0.71 | 0.67 | 0.81 | 0.72 | 0.76 | 0.79 | 0.74 | 0.76 | 0.74 | 0.85 | 0.79 | 0.81 | 0.76 | 0.79 | 0.00 |
| Chemical-I | 0.75 | 0.60 | 0.67 | 0.35 | 0.65 | 0.46 | 0.00 | 0.00 | 0.00 | 0.64 | 0.71 | 0.67 | 0.79 | 0.50 | 0.61 | 0.83 | 0.51 | 0.63 | 0.76 | 0.63 | 0.69 | 0.02 |
| Macro Avg. | 0.78 | 0.73 | 0.75 | 0.58 | 0.62 | 0.58 | 0.41 | 0.46 | 0.42 | 0.75 | 0.73 | 0.74 | 0.74 | 0.70 | 0.71 | 0.76 | 0.75 | 0.74 | 0.79 | 0.73 | 0.76 | 0.01 |

Table 3 shows the results for the two subsets of the two datasets. From the results four things stand out: 1) Augmenting original data with our synthetically generated data always out-performs a model trained with the original data; 2) using **only** synthetically generated data is comparable in performance to the original data in medium labeled data setting (10%) subsets; 3) our synthetically generated data outperforms original data in low labeled data setting (1%) subsets; and 4) our synthetically generated data gives better performance vs two baseline methods: random swap [22] and EntInjection [8].

Our finding that using synthetically generated data can get us a comparable performance to the model trained on real data has an application in making the models trained for downstream tasks like NER, privacy preserving as they are not trained on the real data. This makes it *difficult for the model to leak sensitive data* [11, 2]. Our results show our method of generation can be *quite effective as a data augmentation method in a low labeled data regime*.

### 5.3 Ablation: Generating more text in low resource setting

In the previous results, we only showed the results by generating synthetic text of the same size as the training data. Next, we perform an experiment to see if there is further improvement in the performance as we add more generated text. We take the JNLPBA (1%) dataset, and generate more text using Random Swap and our method. We observe that the results keep improving as measured by the Macro Average $F_1$ score, with augmentation going up to 0.70, and without augmentation going to 0.64 vs baseline at 0.31. Note, we only use the entity mentions found in the JNLPBA (1%) dataset to fill in the entity tags in the generated text. This is more remarkable considering that a model trained on 10x real data for JNLPBA (10%) has a Macro Average $F_1$ score of 0.72. This evidence shows that our model is able to generate text that is similar to the real data.

## 6 Conclusion and Future Work

Synthetic data generation is a promising approach to train large language models in order to deal with scarcity of labeled data, privacy concerns, and data bias. In this work, we study the problem of conditional text generation where the conditions are provided as a list of entities that must appear in the

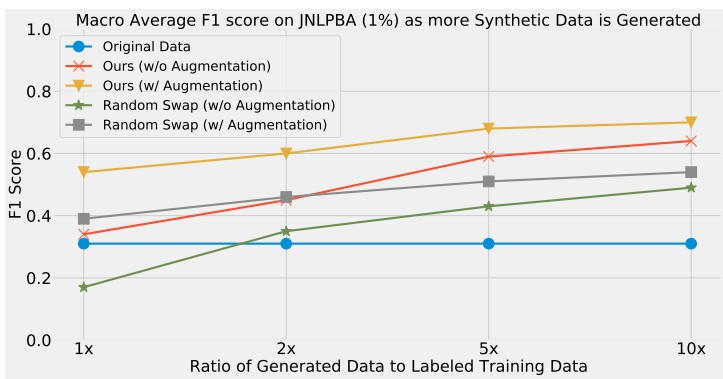

Figure 4: Macro Average F1 score as we augment more generated data to the JNLPBA (1%) dataset.

text in a manner desired by the user. We propose a contextual question and answering approach using pre-trained language models that can generate blocks of text conditioned on the desired entities. We test our generation system on various generation quality metrics as well as on NER tasks. Evaluations show that our proposed method outperforms baselines in terms of both generation quality and NER performance. We also achieve comparable performance relying solely on synthetic data, showing that our proposed architecture can preserve privacy. In future, we will extend this work on more datasets, and explore cross-domain data generation strategies to generate of out-of-distribution data.

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

# A   Appendix

## A.1   Examples of Generated Text

In the section below we shows few examples of generated text by our method and EntInjection [8] method. Our method generates semantically meaningful examples, while EntInjection generates quite repetitive examples. Text highlighted in Green marks the entities.

### A.1.1   Ours

The examples below seem grammatically correct, as was the observation over the entire generated corpus. However, as we randomly insert entity mentions after we generated the entity tags, most of the generated examples are not factual. E.g., DTG is not associated with treatment of blood clotting as generated in the first example. Our goal was not factual correctness but ensuring that the generated data preserves the distribution of the training data, which seems to be the case based on generation metrics and results on NER task.

> The efficacy of DTG in the treatment of impaired blood clotting likewise did not appear to be affected by the rate of administration, although no formal statistical comparisons were made .

> The prevalence rate for death was the most important reason for preference, cited by 67 . 3 % of patients preferring Picloxydine and 54 . 2 % of patients who preferred a p < or = 0 . 001 ) .

> The reduction of acetaminophen at 1 and 4 days after gestation not glomeruli with ataxic movements than control rats .

> The aims of this study were to confirm our previous findings in a separate cohort of patients and to determine the time course of the cardiovascular consequences of stopping sertraline in the expectation that this might shed light on the mechanisms by which the mechanisms by Tamoxifen is being a significant reduction of the activity on the drug causes the sodium associated with cephalothin sodium associated with povidone - iodine is associated withcocaine and inhibition with the use of tuberculosis and area in this effect.

> MR imaging with quantitative diffusion mapping of E4031 ( 0 . g ), p - choloroaniline ) and outcome in organ transplant controls, and / L and the development of blood coagulation by a potential is also more than the development of systolic dysfunction and possibly .

### A.1.2   EntInjection [8]

We observed a lot of repetition in the generated text by EntInjection method. This looping behavior is shown in Example 2 and 3 below. Note, unlike our method, EntInjection has access to the exact same entity mentions as they appear in the training data, having an inherent advantage with this additional information.

> telithromycin - induced bromo tetrahydropyranyladriamycin pituitary carsinom agitation one : a longitudinal study of

> <unk> : The cardiovascular responses to standing and standing . 4 patients were studied in the drug . 4 days . 4 . 4 . 4 . 4 . 4 . 4 . 4 . 4 . 4 . 4 . 4 . 4 . 4 . 4 . 4 . 4 . 4 . 4 . 4 . 4 . 4 . 4 . 4 . 4 . 4 . 4 . 4 . 4 . 4 . 4 . 4 . 4 . 4 . 4 . 4 . 4 . 4 . 4 . 4 . 4 . 4 . 4 . 4 . 4 . 4 . 4 . 4 . 4 . 4 . 4 . 4 . 4 . 4 . 4 . 4 . 4 .

The possibilities that these findings might be the result of non - induced <> is a result of monoamine oxidase or inhibition of monoamine oxidase or inhibition of monoamine oxidase or inhibition of monoamine oxidase or inhibition of monoamine oxidase or inhibition of monoamine oxidase or inhibition of monoamine oxidase or inhibition of monoamine oxidase or inhibition of monoamine oxidase or inhibition of monoamine oxidase or inhibition of monoamine oxidase or inhibition of monoamine oxidase or inhibition of monoamine oxidase or inhibition of monoamine oxidase or inhibition of monoamine oxidase or inhibition of monoamine oxidase or inhibition of monoamine oxidase or inhibition of monoamine oxidase or inhibition of monoamine oxidase or inhibition of monoamine oxidase or inhibition of monoamine oxidase or inhibition of monoamine oxidase or inhibition of monoamine oxidase or inhibition of monoamine oxidase or inhibition of monoamine oxidase or inhibition of monoamine oxidase or inhibition of monoamine

In the study was undertaken to the combination of painful , headache , bleed , which was only induced by epilepticus drug , and bronchitis

Investigation of anti - inflammatory agents are warranted in the caudate nucleus . injection of Allopurinol injection of bacterial collagenase - induced

