# OpenReview forum: "Entity-Controlled Synthetic Text Generation using Contextual Question and Answering with Pre-trained Language Models"
_NeurIPS.cc/2022/Workshop/SyntheticData4ML — Neurips 2022 SyntheticData4ML_

### Official Review · Reviewer_SCJ2 · 2022-10-16
**Entity controlled synthetic text generation via QA by LM**

**Rating:** 7
**Confidence:** 4

**Review:**

The paper proposes a prompt (fig 2) to create training samples for GPT-2 to learn to generate synthetic text data. The generated text shows promising quality and improves downstream NER performance.

In Section 3.1, where do "text samples" come from? Assuming the text samples are real data, how does the method ensure diversity of generated text?

Section 3.1 only considers preceding text blocks as context. Why succeeding text blocks are not considered?

Line 120, left single quotation mark.

Line 128, what does "original training data" refer to?

Line 168, more descriptions about the datasets are needed.

The manuscript needs to uses 2022 NeurIPS template (rather than 2020).

---

### Official Review · Reviewer_5Q6x · 2022-10-18
**Novel idea; empirical results back up the approach**

**Rating:** 8
**Confidence:** 4

**Review:**

Reviewer Summary:
The author(s) address the problem of synthetic data generation and data augmentation for NER tasks. Data augmentation is a heavily studied problem, especially recently since the advent of deep learning and the associated requirements of large training datasets. However, most of these methods are not specifically designed to retain entities for NER. The authors frame entity-controlled text generation as an answer generation task using a pre-trained language model. Unlike typical paraphrasing methods for data augmentation, this method is capable of generating a completely new synthetic data set. Therefore, the approach is not only applicable for data augmentation but also can be used as a generation scheme for producing a synthetic privacy preserving dataset which potentially can also not suffer from some typical biases in available datasets. The authors present empirical evidence that both evaluates the generation quality of their synthetic data, and also its impact on downstream NER performance. They also benchmark their approach against 3 baselines: original dataset, random swap and previous work EntInjection.

Review:
The approach suggested by the authors seems novel and original, to the best of my knowledge. The framing of the synthetic (entity controlled) generation problem as a Context, Question and then Answer generation problem on chunks seemed truly novel.
Their experimental setup, baselines, metrics for evaluation and ablations all seem sound and detailed enough to be convincing. The results performance proved the efficacy of their method. One of the most impressive aspects of this approach was that their synthetically generated dataset was able to outperform original datasets in low resource setting, equal them in medium-resource setting and also outperform 2 baseline methods of random swap and EntInjection. The downstream results when used for data augmentation also clearly show the efficacy of this method.
On the evaluation of generation quality, the automated metrics they use can be misleading. Some form of human evaluation of a fully synthetic dataset, would have been useful, however I understand it can be expensive.
Overall this paper proposes a novel solution to a problem of high practical importance i.e. entity controlled generation and provides rigorous empirical evidence to back up its claim.
I would accept.

---

### Official Review · Reviewer_zBiE · 2022-10-18
**Not properly evaluated in context of previous works; Unclear significant contribution over previous works**

**Rating:** 3
**Confidence:** 4

**Review:**

Pros:
1. The authors provide a finetuned Transformer-based approach to generate NER synthetic data  and an easy way to generate multiple labelled samples for training using an existing labelled dataset.
2. The paper is written clearly and concisely
3. NER performance gains

Cons:
1. Considerable missing literature review. Please see the following papers:
- [MELM: Data Augmentation with Masked Entity Language Modeling for Low-Resource NER](https://aclanthology.org/2022.acl-long.160) (Zhou et al., ACL 2022)
- [Improving Low-Resource Named Entity Recognition via Label-Aware Data Augmentation and Curriculum Denoising](https://aclanthology.org/2021.ccl-1.101) (Wenjing et al., CCL 2021)
- [DAGA: Data Augmentation with a Generation Approach for Low-resource Tagging Tasks](https://aclanthology.org/2020.emnlp-main.488) (Ding et al., EMNLP 2020)
2. Missing baselines - Please provide a comparison with how models like RNNs or GPT-2 using linearized/inline annotated input that have been used in the literature previously to generate synthetic data (example, Ding et al., Zhou et al.) as compared to restructuring the data in a question-answer format and then fine-tuning Transformers to do the same generation.
3. "We clearly observe that our method is lower on all three metrics (Grammaticality, Perplexity, Distinctness-trigram) against the original dataset, which is expected as ours is synthetically generated data"  - Grammaticality is the only metric the authors use to make sure the generated text is of good quality in terms of coherence. Given that it is considerably lower in the synthetic data as compared to the original data, it seems that the generated data while improving NER performance is not mimicing real-world. Thus, the performance increase with just using synthetic data (no gold data) seems spurious. Are there any statistical significance tests that were conducted?

Overall, the only contribution seems to be reformatting the input data in a question-answer format rather than linearlized (annotated inline) format before finetuning a Transformer model that has been prevalent in previous works.  Since comparisons with GPT-2/Transformer-based model trained on input data that is linearized has not been provided by the authors, it is difficult to say whether this work has any improvements over previous works.

---

### Meta-Review · Area_Chair_16iZ · 2022-10-20

**Recommendation:** Accept

**Review:**

Despite a reasonable range of scores, I'm minded to think that this paper clearly presents a novel idea with promising results, and as such would be appropriate for the workshop. If the authors plan to expand on this work in the future, especially at main conferences, I would definitely suggest a more complete handling of related work and experimental comparison.